# Leveraging Diffusion-Based Image Variations for Robust Training on Poisoned Data

**Lukas Struppek**∗
Technical University of Darmstadt

**Martin B. Hentschel**∗
Technical University of Darmstadt

**Clifton Poth**∗
Technical University of Darmstadt

**Dominik Hintersdorf**
Technical University of Darmstadt

**Kristian Kersting**
Technical University of Darmstadt
Centre for Cognitive Science
German Research Center for Artificial Intelligence (DFKI)
Hessian Center for AI (hessian.AI)

## Abstract

Backdoor attacks pose a serious security threat for training neural networks as they surreptitiously introduce hidden functionalities into a model. Such backdoors remain silent during inference on clean inputs, evading detection due to inconspicuous behavior. However, once a specific trigger pattern appears in the input data, the backdoor activates, causing the model to execute its concealed function. Detecting such poisoned samples within vast datasets is virtually impossible through manual inspection. To address this challenge, we propose a novel approach that enables model training on potentially poisoned datasets by utilizing the power of recent diffusion models. Specifically, we create synthetic variations of all training samples, leveraging the inherent resilience of diffusion models to potential trigger patterns in the data. By combining this generative approach with knowledge distillation, we produce student models that maintain their general performance on the task while exhibiting robust resistance to backdoor triggers.

## 1 Introduction

Machine Learning models continually open up new frontiers and turn existing processes upside down. While the general advantage of these models is apparent, their security concerns are often overlooked, potentially resulting in significant threats in practical applications. One prevalent safety and security vulnerability these models face is the threat of backdoor attacks [Gu et al., 2017, Liu et al., 2018]. Backdoor attacks involve training a model on a manipulated dataset, commonly referred to as a poisoned dataset [Barreno et al., 2006], to introduce hidden backdoor functionality. Querying the model with inputs that contain a pre-defined pattern triggers the activation of the backdoor, resulting in behavior unexpected to the user. Detecting these backdoors in a trained model proves challenging because the model hardly exhibits any conspicuous predictions when clean inputs without triggers are fed in [Zhang et al., 2022]. Moreover, identifying poisoned training samples is difficult for humans, as most trigger patterns are designed to be discreet and evade detection by manual dataset inspection. Hence, there is an urgent need for research into innovative and novel defense strategies capable of effectively mitigating the risk posed by backdoor and poisoning attacks.

---

∗Equal Contribution. Contact: lukas.struppek@cs.tu-darmstadt.de

Published at NeurIPS 2023 Workshop on Backdoors in Deep Learning: The Good, the Bad, and the Ugly.

In this work, we introduce a novel defense method based on the latest advancements in diffusion-based image synthesis [Rombach et al., 2022, Xu et al., 2022]. More specifically, our approach first trains a teacher model on a potentially poisoned dataset, following standard training procedures to establish a general understanding of the dataset domain. Subsequently, we leverage a publicly available diffusion model to create synthetic variations of all training samples while preserving the overall image content and domain characteristics. Given that the generative model is highly unlikely to react to triggers within the samples – remember that these triggers are designed to be inconspicuous – they have minimal influence on the content present in the synthetic variations. Finally, we train a separate student model on the synthetic samples, guided by the teacher model. Importantly, unlike related defenses [Li et al., 2021, Pang et al., 2023, Xia et al., 2022], our method requires no additional data at all but relies solely on the original dataset to produce additional synthetic samples. This is a significant improvement, considering the cost and potential privacy concerns associated with acquiring additional labeled data. Our experimental evaluation on large-scale datasets demonstrates that our approach effectively mitigates the influence of poisoned data samples. It almost entirely removes the backdoor behavior from the student model without compromising its utility substantially.

## 2    Background and Related Work

We start by introducing the concept of data poisoning and backdoor attacks together with possible mitigation strategies in Sec. 2.1. We then briefly describe the intuition of diffusion models in Sec. 2.2.

### 2.1    Data Poisoning and Backdoor Attacks

Data poisoning attacks [Barreno et al., 2006] are a class of security attack that manipulates a target model's training process to inject an undesired model behavior for some or all inputs at inference time. In most settings, poisoning attacks are performed by manipulating the model's training data. Given a training set $X_{train} = \{(x_i, y_i)\}$ of data samples $x_i$ with ground-truth labels $y_i$, the adversary adds a relatively small amount of manipulated samples $\widetilde{X} = \{(\widetilde{x}_i, \widetilde{y}_i)\}$ to create a poisoned training set $\widetilde{X}_{train} = X_{train} \cup \widetilde{X}$. Throughout this paper, we mark poisoned datasets and models in formulas with a tilde accent. The target model is then trained on $\widetilde{X}_{train}$, resulting in the poisoned model $\widetilde{M}$. To keep the model manipulation undetected, poisoning attacks generally aim for the trained model to perform comparably to clean models but show some pre-defined behavior in certain use cases.

Backdoor attacks, also known as Trojan attacks, can be interpreted as a special case of targeted poisoning attacks that inject a hidden model behavior that is only activated if some trigger pattern $t$ is present in an input sample $\widetilde{x} = x \oplus t$. We denote the operation of adding a trigger to an input by the $\oplus$ operator. Many backdoor attacks [Gu et al., 2017, Saha et al., 2020, Liu et al., 2018] focus on image classification with the goal of classifying any input $\widetilde{x}$ containing the trigger $t$ as some pre-defined target class $\hat{y}$ that deviates from the sample's ground-truth label. For example, the model classifies all inputs with a white square in a corner as a cat, independently of the actually depicted content. Subsequent lines of work extended backdoor attacks to various other settings beyond image classification [Chen et al., 2021, Zhang et al., 2021, Saha et al., 2022, Struppek et al., 2023].

Defenses against backdoor and poisoning attacks can be grouped into three categories: *data and model analysis*, *backdoor removal*, and *robust model training* [Goldblum et al., 2023]. Approaches from the *data and model analysis* try to identify poisoned data or models but offer no further mitigation strategies to remove backdoors [Paudice et al., 2018, Huang et al., 2020]. This issue is tackled by *backdoor removal* strategies, which try to repair backdoored models after they have been trained on poisoned data without requiring full retraining. And *robust model training* aims to prevent backdoor integration during training in the first place. Our mitigation strategy can be seen as a backdoor removal approach but requires full training of a student model. It also partly belongs to the group of robust model training since we train a second model with knowledge distillation [Hinton et al., 2015] to be robust against backdoor attacks. Previous work also explored variants of knowledge distillation to overcome backdoors [Li et al., 2021, Pang et al., 2023, Xia et al., 2022]. However, unlike our approach, these approaches rely on additional clean data, most of them even require labeled data, and also complicate the training process. We stress that our image variation approach replaces the requirement for additional clean data and can, in principle, be combined with existing approaches. In this work, we investigate the effectiveness of synthetic data independently of other approaches.

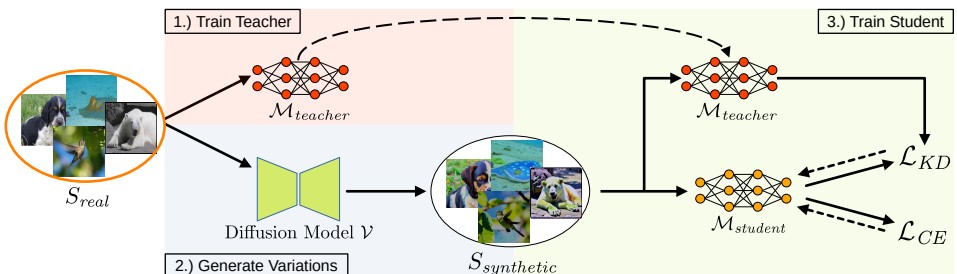

Figure 1: Overview of our defense method. We first train a teacher model $\mathcal{M}_{teacher}$ on the real data $S_{real}$, which might contain poisoned samples. To remove backdoor triggers and other image manipulations, we use a diffusion model $\mathcal{V}$ to generate synthetic variations of all images in the dataset. Finally, we train a student model $\mathcal{M}_{student}$ from scratch on the synthetic data in combination with knowledge distillation to mitigate possible label noise due to dataset poisoning.

## 2.2    Data Synthesis with Diffusion Models

Diffusion-based generative models received much attention from the research community, as well as the general public. Diffusion models [Ho et al., 2020, Song and Ermon, 2020] are trained to denoise images to which random Gaussian noise has been added. The models then learn to revert this diffusion process, allowing them to generate images by denoising samples drawn from a random probability distribution. Recent diffusion models, e.g., Stable Diffusion [Rombach et al., 2022], have shown astonishing performance in the text-guided image generation domain, which generates images based on textual descriptions. Besides textual inputs, related approaches also support conditioning the generation process on already existing images. Versatile Diffusion [Xu et al., 2022] is one representative model that supports the processing of input images to generate synthetic variations of them, among other capabilities like text-to-image synthesis and image captioning. Its pipeline is designed as a unified multi-flow diffusion framework, which enables cross-modal processing of visual and textual contexts and achieves strong results across various domains and tasks. For our experiments, we will use the model's image variation functionality. However, adding textual guidance might be an interesting avenue for future work.

Recent work in the intersection of backdoor attacks and diffusion models aims to integrate backdoors into the diffusion models [Struppek et al., 2023, Chou et al., 2023]. Our research takes another direction and investigates whether general purpose diffusion models can help us to overcome poisoned datasets and prevent the integration of backdoors into models during their training process.

## 3    Methodology

We now describe how we can utilize general-purpose diffusion models to overcome poisoned datasets and related backdoor attacks. Our pipeline, depicted in Fig. 1, consists of three steps: 1.) training a (potentially poisoned) teacher model, 2.) generating image variations of all training samples, and 3.) training a student model on the synthetic data with guidance provided by the teacher model.

**1.) Teacher Model Training.** Our method first trains a teacher model $\mathcal{M}_{teacher}$ on the original dataset $S_{real} = \{(x_i, y_i)\}$, consisting of images $x \in X$ with labels $y \in Y$. Our approach makes no direct assumption about the poisoning grade of the dataset or if the dataset is manipulated in the first place. Training is then done using standard procedures, e.g., optimizing a cross-entropy loss, and no explicit mitigation strategy is applied. Therefore, if trained on poisoned data, the teacher might incorporate the undesired backdoor functionality. We stress that applying robust model training methods to mitigate the integration of backdoors in combination with our diffusion-based defense is an interesting avenue for future research. However, in this work, we focus solely on the potential and effectiveness of synthetic image variations as a defense mechanism.

**2.) Generating Image Variations.** Next, we generate synthetic variations of all training samples in $S_{real}$. To purify sample $x$ of a dataset, we feed it into a diffusion-based image variation model $\mathcal{V}: X \to X$ to generate $k$ synthetic variants $\hat{x} = \mathcal{V}(x)$ of each sample. Importantly, we generate the variations without specifying any class label or additional description. The image generation is solely guided by the image features extracted by the model's encoder. Since the output dimension

of the synthetic samples might differ from the input dimensions, we resize the synthetic samples accordingly to match the size of the original dataset. During our experiments, we set $k = 1$ to keep the dataset size consistent. Importantly, we do not know in advance whether an input sample is a clean or a poisoned sample. Therefore, we generate synthetic variations for all dataset samples.

Due to the nature of diffusion models trained on vast datasets, the generated image variations might render different color statistics compared to the original data, which hurts a model's generalization. To remove dominant and undesirable color casts and improve the student's generalization, we apply a statistical color transfer by adjusting the pixel values of each generated image $\hat{x}$ to obtain a color-adjust sample $\hat{x}'$ matching the statistics of the real image $x$:

$$\hat{x}' = \frac{\hat{x} - \mu(\hat{x})}{\sigma(\hat{x})} \cdot \sigma(x) + \mu(x). \tag{1}$$

Here, $\mu$ and $\sigma$ denote the mean and standard deviation of the pixel values in the real image $x$ and the synthetic variation $\hat{x}$. We then build a new dataset $S_{synthetic} = \{(\hat{x}'_i, y_i)\}$, which shares the same labels with $S_{real}$ but each image is replaced by its color-adjusted synthetic variation. Whereas we find this simple approach to be quite successful, we emphasize that more elaborated color transfer methods might further improve the image statistics.

**3.) Student Model Training.** Finally, we train a separate student model solely on the synthetic dataset. The content fidelity of current image variation models is not always reliable and the depicted content in the synthetic image might differ from the target class. To mitigate this problem and improve the model generalization, we utilize the knowledge incorporated in the model trained on the real data by applying knowledge distillation [Hinton et al., 2015] during training. Knowledge distillation is a common learning paradigm to transfer the knowledge of a usually larger teacher model into a smaller student model. Conceptually, this is done by generating pseudo-labels with the teacher model for the training samples used to update the student model. Let $z_{teacher}$ and $z_{student}$ denote the output logits of the teacher and student model, respectively. Be further $\sigma_j$ the softmax score for the $j$-th class out of a total of $c$ classes, and $\tau$ a temperature term. The knowledge distillation loss is then defined by

$$\mathcal{L}_{KD} = -\tau^2 \sum_{j=1}^{c} \sigma_j \left( \frac{z_{teacher}}{\tau} \right) \cdot \log \sigma_j \left( \frac{z_{student}}{\tau} \right). \tag{2}$$

The total loss consists of $\mathcal{L}_{KD}$ and the standard cross-entropy loss $\mathcal{L}_{CE}$ weighted by factor $\alpha \in [0, 1]$:

$$\mathcal{L} = \alpha \cdot \mathcal{L}_{KD} + (1 - \alpha) \cdot \mathcal{L}_{CE}. \tag{3}$$

Compared to training only on the synthetic data with guidance by the teacher model, knowledge distillation significantly improves the models' prediction accuracy on clean inputs, while also removing the undesired behavior on inputs containing the trigger patterns. We stress that generating image variations is able to remove the triggers from poisoned samples but the assigned labels are still pointing to the wrong class. Here, we exploit that the backdoor of the teacher model is usually no longer triggered on synthetic inputs, therefore, the teacher provides guidance towards the true class which is missing from the label itself.

## 4 Experimental Evaluation

We evaluate our proposed defense mechanism in practical settings. We first introduce the general setup in Sec. 4.1 before discussing our results in Sec. 4.2. Appx. A states more experimental details.

### 4.1 Experimental Setup

**Datasets.** We focus our investigation on high-resolution, real-world datasets and use ImageNette [Howard, 2019], which is a subset of 10 classes from the ImageNet dataset, and a custom ImageNet [Deng et al., 2009] subset with 100 randomly selected classes for our evaluation. All samples are resized to $224 \times 224$ before adding triggers.

**Architectures.** We perform our experiments on the common ResNet-101 [He et al., 2016] architecture. Image variations are generated with Versatile Diffusion [Xu et al., 2022].

**Hyperparameters.** Backdoor attacks are performed in an all-to-one setting, i.e., each triggered backdoor forces the prediction of a single target class, the class with index zero in our case. We

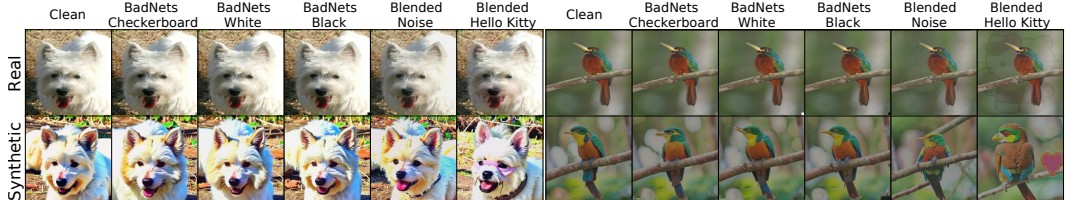

Figure 2: Visual examples of real samples containing triggers and their synthetic variations. While the initial trigger patterns are removed in all variations, some show visible trigger artifacts.

poison $10\%$ of the ImageNette and $1\%$ of the ImageNet-100 training samples, excluding samples from the target class. Knowledge distillation is performed with $\alpha = 0.5$ and $\tau = 5$, but overall, we find the choice of these parameters to only have a minor impact on the results.

**Metrics.** To assess the impact of backdoor attacks on a model, we use two common metrics in line with related work. First, the clean accuracy $Acc_{clean}$, which describes a model's prediction accuracy on the clean test split from the original dataset. And, second, the attack accuracy $Acc_{attack}$, which measures the attack's success by computing the ratio of which samples that contain the trigger are classified as the target class. To compute the attack accuracy, we remove all samples of the target class from the test data and then add the triggers to all remaining samples. Both metrics are computed on a test split which has no sample overlapping with the training data.

**Investigated Backdoor Attacks.** We investigate two common attacks with a total of five variants:

*BadNets* [Gu et al., 2017]: Places fixed trigger patterns in the corners of images. We investigate three trigger variants, namely *Black*, *White*, and *Checkerboard*. The first two options insert a black or white square, respectively, into the images, the last one adds a checkerboard pattern. We set the trigger size to $9 \times 9$ pixels and place the triggers in the bottom right corner of the images.

*Blended* [Chen et al., 2017]: Adds triggers by computing a convex combination between the clean sample, weighted by $1 - \alpha$, and the trigger pattern, weighted by $\alpha$. The trigger pattern either consists of a fixed Gaussian *Noise* pattern or a *Hello Kitty* image. We set the blended ratio to $\alpha = 0.1$ for the noise pattern and to $\alpha = 0.05$ for the *Hello Kitty* trigger.

While both attacks add rather large and visible triggers to samples, more advanced backdoor attacks aim to hide any visual conspicuities and exploit a model's individual processing of samples. By investigating attacks with visible triggers, our experiments build a baseline for the effectiveness of the approach. We expect it to work even better on attacks with less visible trigger patterns since the diffusion model has no incentive to include those patterns in the synthetic image variations.

## 4.2 Diffusion-Based Variations Effectively Mitigate Backdoors

Our experimental results for ResNet-101 models, which are stated in Tab. 1, demonstrate the strong effectiveness of our diffusion-based defense mechanism. In addition, Fig. 2 depicts visual examples of poisoned samples and their corresponding synthetic variations. Training on synthetic ImageNette variations only reduces a model's $Acc_{clean}$ by a single percentage point. At the same time, the $Acc_{attack}$ drops significantly, indicating that the backdoor triggers lose their effectiveness on the student models. The *Blended* attack with the *Hello Kitty* trigger pattern is the only exception here, for which the student model reacts to the trigger in about one-fifth of the cases. Still, attack success is notably lower than for the model directly trained on real poisoned data.

The pattern discussed also holds true for our experiments with 100 ImageNet classes. Again, the student models exhibit strong $Acc_{clean}$ whereas $Acc_{attack}$ drops close to zero. Again, the *Blended* attack with the *Hello Kitty* trigger seems to be robust to our method. Importantly, this attack trigger is much less stealthy than other attacks and adds a large, noticeable pattern to the images that spans all pixels. It is, therefore, little surprising that such strong patterns are also captured by the diffusion models. Hence, the generated image variations frequently contain pink color or flower patches, which may act as new backdoor triggers. However, in practical backdoor scenarios, trigger patterns are designed to be stealthy and usually only affect small parts of images. We therefore expect our approach to work reliably in practice, and even better the less visible trigger patterns are designed.

Table 1: Evaluation results for ResNet-101 models trained on ImageNette and ImageNet-100, respectively. The table compares standard training on real samples to our defense mechanism, which trains student models on synthetic image variations in combination with knowledge distillation. Training on synthetic data only reduces a model's clean accuracy slightly, while most backdoor triggers lose their impact and are ignored by the model.

| | | ImageNette | | | | ImageNet-100 | | | |
|---|---|---|---|---|---|---|---|---|---|
| | | Standard | | Variations + KD | | Standard | | Variations + KD | |
| Attack | Trigger | $Acc_{clean}$ | $Acc_{attack}$ | $Acc_{clean}$ | $Acc_{attack}$ | $Acc_{clean}$ | $Acc_{attack}$ | $Acc_{clean}$ | $Acc_{attack}$ |
| Clean Baseline | | 82.75% | - | 81.07% | - | 74.32% | - | 72.64% | - |
| BadNets | Checkerboard | 80.31% | 99.89% | 79.11% | 1.05% | 74.32% | 99.90% | 71.52% | 0.40% |
| BadNets | White | 80.74% | 91.97% | 80.18% | 1.39% | 75.28% | 95.94% | 72.32% | 0.40% |
| BadNets | Black | 80.51% | 95.70% | 79.82% | 1.50% | 74.22% | 94.40% | 71.26% | 0.67% |
| Blended | Noise | 80.89% | 96.89% | 79.97% | 3.11% | 75.38% | 91.05% | 72.68% | 4.04% |
| Blended | Hello Kitty | 80.76% | 85.36% | 79.49% | 18.80% | 74.48% | 79.98% | 72.24% | 84.34% |

We also provide an extensive ablation and sensitivity analysis in Appx. B. More specifically, we show that training only on synthetic variations without knowledge distillation hurts a model's $Acc_{clean}$ significantly. In addition, we provide results for different trigger sizes, patterns, and colors. Overall, our approach successfully removes the backdoor behavior in almost all cases, except triggers with striking color schemes, e.g., blue and red patches.

## 5 Discussion, Limitations and Future Research

Data poisoning and backdoor attacks pose a serious security risk in practical machine learning applications. Mitigating their impact or defending models during training often requires complex optimization methods and access to trusted labeled datasets. By leveraging a standard diffusion model, we aim to overcome the requirement for additional data and instead generate new samples as variations of existing ones. Our experimental evaluation demonstrates that diffusion-based image variations of poisoned samples significantly reduce the success of backdoor attacks. By using publicly available, general-purpose models like Versatile Diffusion, users can apply our proposed defense strategy directly to a wide range of high-resolution datasets without additional fine-tuning required.

However, our diffusion-based defense method also incorporates some limitations. Generating synthetic variations of images takes slightly less than a second per image. Whereas this time requirement is feasible for small and medium-sized datasets, time can be a constraining factor for vast datasets. Also, for fine-grained tasks like classifications of medical images, a deeper domain understanding of the diffusion model might be required. Fine-tuning a diffusion model on the available data probably improves results, but we leave empirical evaluation open to future research. Given the current speed of developments and breakthroughs in image synthesizing, we expect domain problems to be solved in the near future. Another challenge for diffusion-based image variations is the persisting label noise of the synthetic samples. For example, the synthetic variations of poisoned images depicting dogs labeled as a cat no longer contain the trigger pattern but are still falsely labeled as cats. For datasets with high poisoning rates, this might hurt the student's performance.

An interesting avenue for future research is the integration of sample labels into the image generation process to make the synthetic images more closely related to their assigned labels. We also envision our approach to be applicable in privacy-sensitive domains to create synthetic datasets that could be shared without revealing the private original data.

## 6 Conclusion

We have introduced a novel diffusion-based defense strategy designed to mitigate the impact of poisoned training sets by creating a synthetic surrogate dataset. Coupled with knowledge distillation, our approach trains student models that exhibit robust performance on clean data and significantly reduce susceptibility to backdoor attacks. Through experimental evaluations conducted on high-resolution datasets, we have demonstrated the practical effectiveness of this method. Anticipating continued advancements in image synthesis, we envision the evolving role of diffusion models as a powerful mitigation strategy for data poisoning. With our research, we hope to inspire future research into the application of generative models to combat poisoning and backdoor attacks.

**Reproducibility Statement.** Our source code is publicly available at `https://github.com/LukasStruppek/Robust_Training_on_Poisoned_Samples` to reproduce the experiments and facilitate further analysis.

**Acknowledgments.** This work was supported by the German Ministry of Education and Research (BMBF) within the framework program "Research for Civil Security" of the German Federal Government, project KISTRA (reference no. 13N15343).

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

# A   Experimental Details

In the following, we provide technical details of our experiments to improve reproducibility and eliminate ambiguities.

## A.1   Hard- and Software Details

We performed all our experiments on NVIDIA DGX machines running NVIDIA DGX Server Version 5.2.0 and Ubuntu 20.04.4 LTS. The machines have 1TB of RAM and contain NVIDIA A100-SXM4-40GB GPUs and AMD EPYC 7742 64-core CPUs. We further relied on CUDA 11.4, Python 3.8.10, and PyTorch 2.0.0 with Torchvision 0.15.1 Paszke et al. [2019] for our experiments. We used the model architecture implementations provided by Torchvision. We further provide a Dockerfile together with our code to make the reproduction of our results easier. In addition, all training and attack configuration files are available to reproduce the results stated in this paper.

## A.2   Training Details

We set a constant seed for all experiments to avoid influences due to randomness. All models are initialized with fixed but random weights. We then trained the models for 100 epochs with the Adam optimizer [Kingma and Ba, 2015] and a learning rate of $0.001$. The learning rate was multiplied by factor $0.1$ after $30$ and $50$ epochs, respectively. The batch size was set to $128$. All samples were resized to $224 \times 224$ to build the different datasets. The only augmentation used was random horizontal flipping with probability $p = 0.5$. For training the student models, we used the same hyperparameters. Knowledge distillation was performed with $\alpha = 0.5$ and $\tau = 5$.

## A.3   Generating Image Variations

Synthetic image variations were generated with Versatile Diffusion, available at `https://huggingface.co/docs/diffusers/api/pipelines/versatile_diffusion`. All input samples had the size $224 \times 224$. For each sample, we generated exactly one variation. The number of inference steps was set to $50$ and the guidance scale to $7.5$. The generated images' size is $512 \times 512$ but has been downsized to match the original size of $224 \times 224$.

# B    Additional Experimental Results

We provide additional results for training student models directly on the synthetic image variations without the incorporation of knowledge distillation. Tab. 2 shows the results for ImageNette, Tab. 3 those for ImageNet-100.

Table 2: Evaluation results for ResNet-101 models trained on ImageNette. The table compares standard training on real samples (*Standard*) to our defense mechanism, which trains student models on synthetic image variations in combination with knowledge distillation (*Variations+KD*). The middle columns (*Variations*) show the training results for training on synthetic variations without knowledge distillation applied.

| Attack | Trigger | Standard | | Variations | | Variations + KD | |
|---|---|---|---|---|---|---|---|
| | | $Acc_{clean}$ | $Acc_{attack}$ | $Acc_{clean}$ | $Acc_{attack}$ | $Acc_{clean}$ | $Acc_{attack}$ |
| Clean Baseline | | 82.75% | - | 74.60% | - | 81.07% | - |
| BadNets | Checkerboard | 80.31% | 99.89% | 66.29% | 10.60% | 79.11% | 1.05% |
| BadNets | White | 80.74% | 91.97% | 65.71% | 8.34% | 80.18% | 1.39% |
| BadNets | Black | 80.51% | 95.70% | 68.05% | 23.71% | 79.82% | 1.50% |
| Blended | Noise | 80.89% | 96.89% | 79.97% | 10.94% | 79.97% | 3.11% |
| Blended | Hello Kitty | 80.76% | 85.36% | 70.45% | 4.10% | 79.49% | 18.80% |

Table 3: Evaluation results for ResNet-101 models trained on ImageNet-100. The table compares standard training on real samples (*Standard*) to our defense mechanism, which trains student models on synthetic image variations in combination with knowledge distillation (*Variations+KD*). The middle columns (*Variations*) show the training results for training on synthetic variations without knowledge distillation applied.

| Attack | Trigger | Standard | | Variations | | Variations + KD | |
|---|---|---|---|---|---|---|---|
| | | $Acc_{clean}$ | $Acc_{attack}$ | $Acc_{clean}$ | $Acc_{attack}$ | $Acc_{clean}$ | $Acc_{attack}$ |
| Clean Baseline | | 74.74% | - | 54.02% | - | 72.64% | - |
| BadNets | Checkerboard | 74.32% | 99.90% | 53.22% | 1.29% | 71.52% | 0.40% |
| BadNets | White | 75.28% | 95.94% | 52.46% | 62.89% | 72.32% | 0.40% |
| BadNets | Black | 74.22% | 94.40% | 53.96% | 67.84% | 71.26% | 0.67% |
| Blended | Noise | 75.38% | 91.05% | 53.76% | 16.06% | 72.68% | 4.04% |
| Blended | Hello Kitty | 74.48% | 79.98% | 53.52% | 1.90% | 72.24% | 84.34% |

## B.1 Trigger Sensitivity Analysis

In this section, we conduct a sensitivity analysis regarding the shapes of the backdoor triggers. For the patch-based backdoor attacks, we experiment with various trigger sizes and different color patterns. More specifically, we repeat the experiments on ImageNette and use *yellow* (#ffff00), *red* (#ff0000) and *blue* (#0000ff) instead of black and white as trigger colors. The results stated in Tab. 4 show that our defense method works well for most color patterns. However, for *red* and *blue* color patches, the method is not able to completely remove the backdoor. Since both color patches are rather striking, the diffusion model tends to integrate similar patches in its generated variations. We also repeat the experiments using the standard black-and-white checkerboard pattern different trigger sizes from the range of 3 up to 27 pixel length. Our results in Tab. 5 demonstrate that the approach even works for larger trigger sizes.

Table 4: Evaluation results for ResNet-101 models trained on ImageNette using different color patches as backdoor triggers. In addition to the black and white triggers, we also evaluated on *yellow*, *red*, and *blue* triggers, as well as checkerboard patterns using combinations of these three colors.

| Attack | Trigger | Standard | | Variations | | Variations + KD | |
|---|---|---|---|---|---|---|---|
| | | $Acc_{clean}$ | $Acc_{attack}$ | $Acc_{clean}$ | $Acc_{attack}$ | $Acc_{clean}$ | $Acc_{attack}$ |
| Clean Baseline | | 82.75% | - | 74.60% | - | 81.07% | - |
| BadNets | Checkerboard | 80.31% | 99.89% | 66.29% | 10.60% | 79.11% | 1.05% |
| BadNets | White | 80.74% | 91.97% | 65.71% | 8.34% | 80.18% | 1.39% |
| BadNets | Black | 80.51% | 95.70% | 68.05% | 23.71% | 79.82% | 1.50% |
| BadNets | Yellow | 81.45% | 99.94% | 69.07% | 96.81% | 80.13% | 1.36% |
| BadNets | Red | 82.06% | 99.86% | 70.75% | 98.87% | 79.95% | 24.14% |
| BadNets | Blue | 82.09% | 99.89% | 71.62% | 98.64% | 80.08% | 97.51% |
| BadNets | Yellow & Red | 79.85% | 99.55% | 67.75% | 31.57% | 79.67% | 1.41% |
| BadNets | Yellow & Blue | 81.30% | 99.72% | 67.16% | 7.77% | 79.82% | 1.14% |
| BadNets | Red & Blue | 80.43% | 100.00% | 70.90% | 91.94% | 79.90% | 1.33% |

Table 5: Evaluation results for ResNet-101 models trained on ImageNette using the black-and-white checkerboard triggers of different sizes. All other hyperparameters are kept the same.

| Trigger | Standard | | Variations | | Variations + KD | |
|---|---|---|---|---|---|---|
| | $Acc_{clean}$ | $Acc_{attack}$ | $Acc_{clean}$ | $Acc_{attack}$ | $Acc_{clean}$ | $Acc_{attack}$ |
| Checkerboard ($9 \times 9$) | 80.31% | 99.89% | 66.29% | 10.60% | 79.11% | 1.05% |
| Checkerboard ($13 \times 13$) | 81.78% | 99.92% | 67.24% | 7.35% | 80.69% | 0.88% |
| Checkerboard ($17 \times 17$) | 81.02% | 99.89% | 69.94% | 11.45% | 79.82% | 0.99% |
| Checkerboard ($21 \times 21$) | 82.78% | 100.00% | 69.32% | 14.73% | 81.35% | 0.73% |
| Checkerboard ($25 \times 25$) | 82.78% | 100.00% | 68.43% | 6.47% | 80.97% | 0.82% |

## B.2 Ablation Study

We further investigate the impact of the different parts of our approach by training a model directly on the synthetic variations without knowledge distillation. In addition, we also train a student model using knowledge distillation but replaced the synthetic variations by the original training data under strong augmentations. Specifically, we apply color jittering ($1\pm0.4$ brightness, contrast and saturation; $\pm0.1$ hue), random rotation ($\pm20$ degrees), random affine transformation ($\pm0.1$ translation, $1 \pm 0.1$ scaling), and random horizontal flipping ($50\%$ probability). The results are stated in Tab. 6. We stress that the random affine transformations destroy the patch-based triggers, which are always located on the bottom right corner. Consequently, the resulting attack accuracy is quite low but might be misleading.

Table 6: Evaluation results for ResNet-101 models trained on ImageNette. The table compares training with knowledge distillation and poisoned training samples under strong augmentation (*Augmentations + KD*), training only on synthetic image variations (*Variations*), and training on synthetic image variations in combination with knowledge distillation (*Variations+KD*).

| Attack | Trigger | Augmentations + KD | | Variations | | Variations + KD | |
|---|---|---|---|---|---|---|---|
| | | $Acc_{clean}$ | $Acc_{attack}$ | $Acc_{clean}$ | $Acc_{attack}$ | $Acc_{clean}$ | $Acc_{attack}$ |
| BadNets | Checkerboard | 80.61% | 2.66% | 66.29% | 10.60% | 79.11% | 1.05% |
| BadNets | White | 81.81% | 2.63% | 65.71% | 8.34% | 80.18% | 1.39% |
| BadNets | Black | 81.38% | 63.68% | 68.05% | 23.71% | 79.82% | 1.50% |
| Blended | Noise | 81.63% | 94.35% | 79.97% | 10.94% | 79.97% | 3.11% |
| Blended | Hello Kitty | 82.62% | 68.00% | 70.45% | 4.10% | 79.49% | 18.80% |

