# OpenReview forum: "Leveraging Diffusion-Based Image Variations for Robust Training on Poisoned Data"
_NeurIPS.cc/2023/Workshop/BUGS — NeurIPS 2023 BUGS Poster_

### Official Review · Reviewer_qdvu · 2023-10-24
**Proposed defense is shown to maintain clean accuracy while reducing backdoor attack success rates**

**Rating:** 7
**Confidence:** 4

**Review:**

## Originality and significance
Propose using synthetic variations from a diffusion model as a train-time augmentation within a student-teacher framework.

## Clarity
- In the experimental setup, the authors clearly state their hypothesis for the experiments they perform: they expect their method to work best “on attacks with less visible trigger patterns”. Even though this intuitive hypothesis turns out not to be the case, I appreciated reading a clear hypothesis before results. It made the results more interesting.
- Table 1 is easy to read and see that Variations+KD reduces attack success rates significantly while maintaining clean accuracy.

## Quality
- To improve the work, it would be nice to see a larger array of potential triggers. For highly visible triggers, checkerboard/white/black can be expanded to include colored grids or larger than 9x9 trigger sizes. For blended triggers, it would be interesting to include images that are not so recognizable/identifiable like Hello Kitty: maybe a synthetic diagonal pattern, checkerboard, or other textured pattern. Adding these additional triggers would add weight to the current results and also potentially answer whether Blended Hello Kitty is the best attack. For example, if the blended checkerboard doesn’t work as well as Blended Hello Kitty, one could begin to conclude that the synthetically generated images are beginning to poison the student model.
- I am a bit confused on how a potentially poisoned teacher model would be useful for training a student model. What happens if we remove the student-teacher framework and train the student model normally on the synthetic variations? This ablation would help in seeing the significance of the full proposal.

Minor:
Based on Section 3, in generating image variations, it appears that synthetic variants can be generated without labels? Does this mean the diffusion model inherently classifies the input and generates a related sample? It would be helpful to explicitly state this. It’s hard to know what would happen if the input image is hard to recognize, or if the image contains a mix of different classes.

## Summary
Diffusion models’ representations of natural images make them a good choice for mapping a backdoor/poison/corrupted image to a more natural image. The proposed defense is shown to maintain clean accuracy while reducing backdoor attack success rates. The workshop would benefit from the presentation of this work. My main suggestion would be to include the ablation on how performance changes if the student-teacher framework is removed, and we train a RN-101 normally on the synthetic variations. This would tell us the importance of synthetic variations.

---

### Official Review · Reviewer_g7yC · 2023-10-25

**Rating:** 6
**Confidence:** 3

**Review:**

Summary - The main idea behind this paper is to train a teacher model and a diffusion model on poisoned data. Then, synthetic samples generated from the diffusion model are used, and a student model is trained using a knowledge distillation framework. The student model is shown to have high accuracy on benign images and exhibits robustness to backdoor triggers. The experimental results are shown on 2 datasets (ImageNette and ImageNet-100), with 5 different poison variants of BadNets, and blended images.

Areas of Improvement -
1. It would be good to show ablation, regarding how much knowledge distillation v/s diffusion model helps this framework.
2. Another important ablation would be to replace the diffusion model, with strong data augmentations such as affine transformations, color jitter etc.
3. The results regarding Blended Hello Kitty for ImageNet-100 are surprising. It would be interesting to know, why the attack is *more successful* for Imagnet-100.

The approach also has a few obvious drawbacks (which the paper discusses) such as the high cost of training a diffusion model, and the requirement of a large dataset to train the same. The results seem interesting to discuss at the workshop.

Minor -
L22 - Backdoors exhibit no conspicuous behavior -> add citations for relevant works that claim this.
L61 - the a -> fix

---

### Decision · Program_Chairs · 2023-10-28

**Decision:**

Accept (Poster)

**Comment:**

The authors utilize a teacher model, diffusion model, and knowledge distillation to defend against backdoor attacks. Specifically, they introduce a novel use of synthetic variations from a diffusion model within a student-teacher framework. The authors should explore a broader range of triggers and conduct an ablation study as suggested by the reviewers.